# Chemosensitivity of 3D Pancreatic Cancer Organoids Is Not Affected by Transformation to 2D Culture or Switch to Physiological Culture Medium

**DOI:** 10.3390/cancers14225617

**Published:** 2022-11-16

**Authors:** Vincent Gassl, Merel R. Aberle, Bas Boonen, Rianne D. W. Vaes, Steven W. M. Olde Damink, Sander S. Rensen

**Affiliations:** 1Department of Surgery, Maastricht University Medical Centre, 6229 HX Maastricht, The Netherlands; 2NUTRIM-School of Nutrition and Translational Research in Metabolism, Maastricht University, 6229 ER Maastricht, The Netherlands; 3Department of Visceral and Transplantation Surgery, RWTH Aachen University, 52074 Aachen, Germany

**Keywords:** organoids, primary cell culture, chemosensitivity, physiological cell culture medium, pancreatic cancer

## Abstract

**Simple Summary:**

Organoids are increasingly used to investigate patient-specific drug responsiveness since they are thought to be more representative of a patient’s tumor than two-dimensional primary cell cultures. Furthermore, cell culture media that mimic physiological nutrient concentrations have been suggested to improve chemotherapy screens of cultured cells. As both come with increased costs and complexity, we investigated the response of two patient-derived pancreatic cancer organoids (PANCO09b, PANCO11b) growing as 3D organoids versus 2D transformed cell cultures in either conventional or physiological media towards five chemotherapeutics (gemcitabine, paclitaxel, SN-38, 5-fluorouacil, and oxaliplatin). Both patient-derived pancreatic cancer cell cultures showed similar drug-responses when cultured in 3D compared to 2D, as well as upon culture in physiological versus conventional culture media, except for higher sensitivity towards SN-38 when PANCO11b was cultured in 2D or in physiological media. These data show that drug-responsiveness of primary pancreatic cancer cells is not majorly impacted by culture conditions.

**Abstract:**

Organoids are increasingly used to investigate patient-specific drug responsiveness, but organoid culture is complex and expensive, and carried out in rich, non-physiological media. We investigated reproducibility of drug-responsiveness of primary cell cultures in 2D versus 3D and in conventional versus physiological cell culture medium. 3D pancreatic ductal adenocarcinoma organoid cultures PANCO09b and PANCO11b were converted to primary cell cultures growing in 2D. Transformed 2D cultures were grown in physiological Plasmax medium or Advanced-DMEM/F12. Sensitivity towards gemcitabine, paclitaxel, SN-38, 5-fluorouacil, and oxaliplatin was investigated by cell viability assays. Growth rates of corresponding 2D and 3D cultures were comparable. PANCO09b had a shorter doubling time in physiological media. Chemosensitivity of PANCO09b and PANCO11b grown in 2D or 3D was similar, except for SN-38, to which PANCO11b cultured in 3D was more sensitive (2D: 8.2 ×10^−3^ ± 2.3 ×10^−3^ vs. 3D: 1.1 ×10^−3^ ± 0.6 ×10^−3^, *p* = 0.027). PANCO09b and PANCO11b showed no major differences in chemosensitivity when cultured in physiological compared to conventional media, although PANCO11b was more sensitive to SN-38 in physiological media (9.8 × 10^−3^ ± 0.7 × 10^−3^ vs. 5.2 × 10^−3^ ± 1.8 × 10^−3^, *p* = 0.015). Collectively, these data indicate that the chemosensitivity of organoids is not affected by culture medium composition or culture dimensions. This implies that organoid-based drug screens can be simplified to become more cost-effective.

## 1. Introduction

Pancreatic cancer is highly lethal, with a five-year survival rate of only 9% [1]. As surgical resection is not feasible in 80% of the cases, chemotherapy is the most common treatment. Whilst some years ago gemcitabine monotherapy was the gold standard, combining gemcitabine with nab-paclitaxel has been shown to prolong survival by approximately 2 months [2]. Nowadays, the most frequently administered chemotherapy is called FOLFIRINOX, and consists of four chemotherapeutic agents, i.e., oxaliplatin, irinotecan, 5-fluorouracil (5-FU), and folinic acid. On average, the FOLFIRINOX regimen achieved 4–5 months longer survival compared to gemcitabine monotherapy [2,3]. However, as patients respond differently to chemotherapy, a model to predict the most effective drugs may result in longer lifespan and less side effects.

In recent years, researchers have made important progress in predicting the patients’ response to chemotherapeutics by using patient derived organoids (PDOs). Organoids are primary cell cultures that can be generated from resected tumors. They grow in a basement membrane extract (BME) which allows them to expand 3-dimensionally in an organ-like structure, and they maintain heterogeneity in terms of cellular and molecular composition ex vivo [4,5,6]. Previously, we showed that pancreatic PDOs retain defining pathophysiological features of the in vivo tumor [7], and Tiriac et al. demonstrated that they can be used to predict patient’s chemotherapy responses as well as development of chemotherapy resistance [8].

While organoids are superior in mimicking in vivo conditions as compared to conventional 2-dimensional cell cultures, their culturing is more time-consuming, highly expensive, and reproducing results is more difficult [9]. As a result, there is high interest in the development of simplified patient-derived culture systems that reliably model drug-responses. Recently, bladder cancer organoids that were transformed into an organoid-derived monolayer were shown to maintain marker expression patterns and drug responsiveness [10]. This approach combines the high efficiency with which PDOs can be established [11] with the application of conventional techniques used for analysis of 2D cultures grown on plastic tissue culture surfaces.

Another recent development in cancer cell culturing is the use of physiological cell culture media like Plasmax and human-plasma like medium (HPLM). These physiological media aim to mimic the nutrient concentrations of human plasma, which are substantially different from the concentrations found in conventional cell culture media like DMEM or RPMI [12,13] (Appendix A). Since nutrient availability profoundly affects cellular metabolism, nutrient concentrations in cell culture media influence cellular drug responses [14,15]. For example, Cantor et al. recently showed that the acute myeloid leukemia cell line NOMO1 had a 5–8 fold decreased sensitivity to 5-FU when grown in media containing uric acid at physiological concentrations compared to conventional media [13]. Yet, organoid cultures are regularly cultured in a standardized medium (advanced DMEM/F-12) designed to support fast growth, but not to mimic in vivo conditions.

Here, we investigated the chemosensitivity of pancreatic tumor PDOs grown in 3D and their corresponding 2D transformed cell cultures in physiological media and conventional culture media.

## 2. Materials and Methods

For this study, we used PANCO09b and PANCO11b organoids from patients with pancreatic ductal adenocarcinoma that were previously established in our laboratory [7].

### 2.1. Maintenance and Passaging of 3D Organoids

Organoids were maintained in Tumor 2 (T2) medium consisting of adv.DMEM/F-12 (Gibco (Waltham, MA, USA), cat. no. 12634010), (Appendix A) supplemented with 1% HEPES (Gibco, cat. no. 15630-080), 1% Glutamax (Gibco, cat. no. 35050-061), and 1% P/S (adv. DMEM/F12+++) as basal medium and additional components added as shown in Appendix A. Medium was freshly made and stored for max. 2 weeks at 4 °C. 3D organoids were cultured in domes of Geltrex LDEV-Free Reduced Growth Factor.

Basement Membrane Matrix (Gibco, Cat. No. 1413202) in 24-wells non-tissue culture (NTC) treated plates (Eppendorf (Hamburg, Germany), cat. no. 0030722116) and passaged every 7–10 days. For passaging, domes were dissolved in ice-cold adv.DMEM/F12+++, and the suspension was collected and centrifuged for 5 min at 300× *g* at 4 °C. Subsequently, supernatant was aspirated and the organoids were mechanically sheared through a narrowed glass Pasteur pipette. After adding cold adv.DMEM/F12+++, organoids were centrifuged for 5 min at 300× *g* at 4 °C. Supernatant was aspirated and ice-cold Geltrex was added. Organoids were seeded in 50 µL/well Geltrex separated in 3 domes in a NTC 24-well plate. After domes were solidified in the incubator at 37 °C and 5% CO_2_, 500 µL T2 media was added. Images were taken with an Incucyte Live-Cell Imaging system (Sartorius, Göttingen, Germany) at 4× magnification.

If cells were used for RNA isolation, supernatant was aspirated from the well and TripLE (ThermoFisher (Waltham, MA, USA), cat. no. 12605010) was added, followed by homogenization via pipetting, and incubation at room temperature for 5 min. The cell homogenate was stored at −80 °C until RNA isolation. All experiments were performed before passage 30.

### 2.2. Transformation of 3D Organoids into 2D Cell Cultures

Organoids were collected, centrifuged, and supernatant including Geltrex was aspirated as described above. Subsequently, the pellet was resuspended in 2 mL TrypLE and incubated for 5–10 min. on a rocker at 37 °C. The cell suspension was pipetted up and down several times. After adding 10 mL pre-warmed adv.DMEM/F-12+++, the cell suspension was centrifuged at 300× *g* for 5 min. Next, supernatant was aspirated and the pellet was resuspended in 2D Tumor 2 (2DT2) medium, consisting of 50:50 (*v*/*v*) T2 and adv.DMEM/F12+++ (Appendix A). Four wells of 3D organoids were used to seed 3–4 wells of 2D organoids into 12-wells tissue-culture treated plates (Eppendorf, cat. no. 0030721110). After seeding, plates were cultured at 37 °C and 5% CO_2_.

### 2.3. Maintenance and Passaging of 2D Transformed Cell Cultures

2D transformed cell cultures were maintained in 2DT2 media in 12-wells tissue-culture treated plates in an incubator at 37 °C and 5% CO_2_. Medium was refreshed every 2–3 days. Cells were passaged when they reached 80% confluency. Cells were covered in 400 µL TripLE and incubated at 37 °C for 5–10 min. Subsequently, cells were collected in a 10 mL tube and centrifuged at 350× *g* for 5 min. Afterwards, TrypLE was aspirated and cells were resuspended and counted using a counting chamber under the brightfield microscope before dilution for passaging (1 × 10^5^ cells/well of a 12-wells plate) or experimental use. If cells were needed for RNA isolation, 1 mL of TRI Reagent was added per well. Lysed cell solution was stored at −80 °C until RNA isolation.

### 2.4. Switching from Conventional to Physiological Culture Medium

To grow 2D transformed cell cultures in physiological medium, 2DT2 medium was prepared with Plasmax [16] (2DPlxT2, will be referred to by its basal media, Plasmax) supplemented with 1% HEPES (Gibco, cat. no. 15630-080), 1% Glutamax (Gibco, cat. no. 35050-061), and 1% P/S (Plasmax+++) as basal media instead of adv.DMEM/F12+++ (Appendix A). Cells growing in 2DT2 were trypsinized and collected as previously described. Subsequently, the pellet was resuspended in 2DPlxT2 and plated as mentioned before. Cells were allowed to adapt to the new environment for 2 passages. Maintenance, passaging, and RNA isolation was done as described above.

### 2.5. Chemosensitivity Assay

Organoid-derived 2D cell cultures were trypsinized and counted as described above. 3D organoids were collected in ice cold adv.DMEM/F12+++ and centrifuged at 200× *g* for 5 min at 4 °C. After supernatant was aspirated, the pellet was resuspended in 1–2 mL of TriplE + RhoKinase Inhibitor (1:1000) and incubated on a shaker at 37 °C until organoids were dissociated into single cells. 3 mL of adv.DMEM/F12+++ was added and the cell suspension was centrifuged at 200× *g* for 5 min at 4 °C. Subsequently, supernatant was aspirated, 1ml adv.DMEM/F12+++ added, and cells were counted using a counting chamber under the brightfield microscope. Single cells were plated on a 384-wells plate (Greiner, cat. no. 781098) at densities of 500 cells/well (3D organoid-derived single cells, 2D culture derived single cells) or 3000 cells/well (only 2D culture derived single cells) in 30 µL medium/well. After cells settled, plates were cultured at 37 °C and 5% CO_2_ in the Incucyte to assess morphology and confluency.

After one day, chemotherapeutics were added using a Tecan D300e digital dispenser (Tecan, Männerdorf, Switzerland) in triplicate. Cells were treated with 14 different concentrations where gemcitabine, paclitaxel, and SN-38 titrations ranged from 1.08 × 10^−5^–2 µM, whilst 5-FU and oxaliplatin titrations ranged from 0.013–50 µM. The negative control was exposed to 0.5% dimethyl sulfoxide (DMSO), and all wells were normalized to this condition. Plates were cultured at 37 °C and 5% CO_2_. After 5 days of treatment, cell viability was assessed using CellTiterGlo (Promega, cat. no. G924B) following the manufacturer’s instructions. Luminescence was measured with the Tecan Spark M10 multimode plate reader.

### 2.6. Quality Control

To assess the quality of the chemo-sensitivity assay, strictly standardized mean difference (*SSMD*) and Z-scores were calculated using Formulas (1) and (2), respectively.
(1)$$SSMD=\frac{\bar{NC}-\bar{PC}}{\sqrt{SD{\left(NC\right)}^{2}+PC{\left(NC\right)}^{2}}}$$
(2)$$Z-score=1-\left(\frac{(\left(3\times SD\left(NC\right)\right)+\left(3\times SD\left(PC\right)\right)}{\left(\bar{NC}-\bar{PC}\right)}\right)$$

*NC*, *PC*, and *SD* stand for negative control, positive control, and standard deviation, respectively. To assure quality of data, the *SSMD* needed to be above 3 and Z-score higher than 0.4.

### 2.7. Dose–Response Curves

Dose–response curves were generated and analyzed using GraphPad Prism 9 (San Diego, CA, USA). The mean luminescence of the control (0.5% DMSO) was set to 100% cell viability. The logarithm of administered drug concentration was plotted against the corresponding cell viability. The graph was fitted using a nonlinear regression curve described by Formula (3) (in Graphpad: log(inhibitor) vs. response—variable slope (four parameter)).
(3)$$y=bottom+\frac{\left(top-bottom\right)}{\left(1+10^{\left(\left(logIC50-x\right)\times HillSlope\right)}\right)}$$

The relative inhibitory concentration 50 (*IC*_50_) value was obtained using logarithmic regression and determined as drug concentration at which half-maximal effect was obtained.

### 2.8. Doubling Time

Doubling time of cells was assessed during the chemosensitivity assays using untreated cells. Luminescence was measured at the start and at the end of the treatment. Doubling time was calculated using Formula (4).
(4)doubling time=duration×ln(2)ln(luminescence(end))ln(luminescence(initial))

Growth rate/day was calculated using Formula (5).
(5)growth rate=ln(luminescence(end))ln(luminescence(initial))duration

### 2.9. Growth Rate 50

Dose–response growth rate 50 (GR_50_) values were calculated using the online tool grcalculator.org, as explained by Clark et al. [17,18].

### 2.10. Proliferation Assay

2D transformed cell cultures were passaged and counted as described above. The cell suspension was diluted and 30,000 cells/well were plated in triplicate in a 48-wells tissue-culture treated plate (Eppendorf, cat. no. 0030723112) in 0.3 mL medium. After cells settled, the plate was put in the Incucyte Live-Cell Imaging system at 37 °C and 5% CO_2_ to monitor confluency over time. Images were taken every 2 h with 9 images/well at 10× magnification. The first image was taken approximately 2 h after plating. The assay was stopped when the stationary phase of the curve was reached.

### 2.11. RNA Isolation

RNA isolation was performed according to the TRI Reagent® Solution manufacturer‘s protocol (ThermoFisher). Cells were homogenized with TRI Reagent and transferred to a 1.5 mL Eppendorf tube. After 5 min. incubation time at room temperature, 200 µL chloroform per 1 mL TRI Reagent was added. Samples were incubated for 15 min. followed by a 15 min. centrifugation at 15,000 rpm at 4 °C. After transferring the aqueous phase (300–400 µL) to a new tube, isopropanol was added at a 1:1 ratio. To initiate precipitation, samples were vortexed and incubated for 10 min. at room temperature, followed by centrifugation at 15,000 rpm for 15 min. at 4 °C. Subsequently, the supernatant was discarded and the RNA pellet was washed twice in 75% ethanol in nuclease free water, followed by 15 min. centrifugation at 15,000 rpm. After decanting, the pellet was dried at 40 °C for 5 min. and resuspended in 10 µL nuclease free-water. RNA was stored at −80 °C until further use.

### 2.12. RNA Sequencing

RNA was isolated from 3D organoids and 2D transformed cell cultures as described above. RNA was sent to Novogene Co., Ltd. (Cambridge, UK) for quality control and mRNA sequencing. Quality and quantity were checked by Novogene using gel electrophoresis (Bioanalyzer Agilent 2100) and Nanodrop. RIN values had to be >8.0 to continue to library preparation. Preparation of RNA library and transcriptome sequencing was conducted by Novogene Co., Ltd. (Oxford, UK). In short, mRNA was purified from total RNA using poly-T oligo-attached magnetic beads. After fragmentation, the first strand cDNA was synthesized using random hexamer primers followed by second strand cDNA synthesis. The library was ready after end repair, A-tailing, adapter ligation, size selection, amplification, and purification, and was checked with Qubit and real-time PCR for quantification and bioanalyzer for size distribution. Quantified libraries were pooled and sequenced on the Illumina NovaSeq 6000 with a 2 × 150 bp high output.

Raw data (Fastq format) was obtained and datasets were trimmed using Fastp to remove adapters, poly-N and low-quality reads from the raw data, and aligned in single-end format to the Ensembl reference genome (GRCh38, release 104) using STAR and quantified using RSEM.

### 2.13. Statistics

Graphical presentation and statistical analysis were performed using GraphPad Prism for macOS version 9.0.0. All data are represented as mean or median ± SD. Time-point related statistical analysis of the proliferation curves was done using two-way ANOVA with Sidak’s correction for multiple comparisons. The Mann–Whitney U test was used to compare differences between two independent samples. Comparison of *IC*_50_ and GR_50_ values was done using unpaired *t*-tests. Statistical significance was assumed when *p* < 0.05. Statistical analysis of RNA sequencing data was done using R-studio version 2022.07.0 for macOS using the packages BiocManager, DESeq2, ggplot2, dplyr, ggpubr, EnhancedVolcano. Genes with Benjamin-Hochberg adjusted *p*-value < 0.05 were considered differentially expressed.

## 3. Results

### 3.1. Establishing 2D Transformed Cell Cultures from 3D Organoids

PANCO09b and PANCO11b organoids both grew in evenly round to oval shaped structures with a lumen (Figure 1A). PANCO09b organoids grew bigger than PANCO11b, reaching a maximal diameter of approximately 400 µm whilst PANCO11b organoids reached a diameter of maximally 250 µm. Both PANCO09b and PANCO11b were successfully transformed to 2D cell cultures with cells readily attaching to the surface of the wells (Figure 1A). The growth rate/day as well as the doubling time in hours did not differ significantly between 3D and 2D PANCO09b (Figure 1B, Appendix A, *p* = 0.786 and *p* = 0.936, respectively) or PANCO11b cultures (*p* = 0.298 and *p* = 0.305, respectively).

To investigate the impact of the 2D versus 3D culture conditions on gene expression patterns, we performed RNA sequencing. There were only 31 differentially expressed genes (Figure 1C, Appendix A), with the most differentially expressed gene being an RNA gene, followed by Regulator of Solute Carrier 1 (RSC1A1, ENSG00000263244, adj. *p* = 2.15 × 10^−9^), which was upregulated in 3D compared to 2D cultures. Furthermore, two cell-cycle related genes showed differential expression (CDKL3 and MYCN) [19,20,21]. These data show that the transformation of the 3D organoid cultures to 2D transformed cell cultures was successful and did not majorly impact core cellular characteristics.

### 3.2. Similar Chemosensitivity of 3D Organoids and Corresponding 2D Transformed Cell Cultures

To compare the chemosensitivity of 3D organoid cultures and their corresponding 2D counterparts, three separate drug-responsiveness assays of PANCO09b and PANCO11b cells grown in adv.DMEM/F-12 were conducted (Figure 2, Appendix A). Dose–response curves of PANCO09b and PANCO11b were highly reproducible for all three experiments in both 2D and 3D (Figure 2A,B).

None of the drugs reduced cell viability by 100% at maximal dose. Using the dose–response curves, the *IC*_50_ of the two PDO cultures was determined (Figure 2C). The *IC*_50_ values of gemcitabine, paclitaxel, SN-38, 5-FU, and oxaliplatin were not different for PANCO09b cells grown in 2D versus 3D. PANCO11b grown in 3D showed significantly higher sensitivity to SN-38 (*p* = 0.027) than the corresponding 2D culture, whilst no differences were observed for gemcitabine, paclitaxel, 5-FU, and oxaliplatin.

All of the administered drugs interfere in DNA-synthesis/replication during cell division. As a consequence, the effectiveness of the drug is associated with the cell division cycle [22]. To correct for the resulting impact on growth rate, the GR_50_ was calculated. GR_50_ is an advanced metric that considers the growth rate as a confounder for drug sensitivity calculations and defines the drug concentration needed to inhibit growth by 50% [17]. The GR_50_ values of gemcitabine, paclitaxel, SN-38, 5-FU, and oxaliplatin did not differ for PANCO09b grown in 2D versus 3D (Figure 2D, Appendix A). PANCO11b showed higher sensitivity to SN-38 in 3D than in 2D culture (*p* = 0.0004), but similar drug responsiveness towards gemcitabine, paclitaxel, 5-FU, and oxaliplatin.

### 3.3. Impact of Physiological Cell Culture Medium on Proliferation and Morphology of 2D Transformed Cell Cultures

Next, the effect of culture medium composition on PANCO organoids grown in 2D was assessed by investigating morphology and proliferation rates in adv.DMEM/F-12 or in Plasmax physiological cell culture medium. After plating the 2D cultures, negligible growth was detected by live cell imaging during the first two days (first data-point is shown at 24 h, Figure 3A). Although growth rate differed, a logistic growth curve was seen for PANCO09b grown in both Plasmax and adv.DMEM/F-12. The density of PANCO09b grown in Plasmax compared to adv.DMEM/F-12 was significantly lower from 96h after seeding until cells reached confluency after 216 h in both conditions when assessed by live cell imaging. As a result, PANCO09b cells grown in adv.DMEM/F-12 reached almost 90% confluency after 144 h, whilst cells grown in Plasmax reached confluency of 90% only after 212 h. Heterogeneity of PANCO09b 2D cultures was maintained in both adv.DMEM/F-12 and Plasmax, as cells were of different shapes and sizes (Figure 3B). On day three, cell colonies rather than single cells were seen in both conditions. However, when 40–50% confluency was reached after approximately five to six days, cells started growing on top of each other, which was more prominent in Plasmax than in adv.DMEM/F-12. This growth habit increased more over time. These areas are highlighted with red boxes in Figure 3A.

Growth curves of PANCO11b grown in adv.DMEM/F-12 and Plasmax differed significantly from 126–218 h after plating, although the difference was small as compared to PANCO09b (Figure 3A). On day four, 2-dimensonally growing PANCO11b cells formed colonies in both culture media instead of being distributed as single cells (Figure 3C). After seven days, these clusters merged in both conditions. Heterogeneity of the cell cultures was evident, with cells of different shapes and sizes growing along each other. In some areas, cells were rather thin and elongated, while in other, less dense areas, cells were oval and bigger. After ten days, cells reached 100% confluency in both adv.DMEM/F-12 and Plasmax. In both conditions, we observed cells growing on top of each other, which was slightly more prominent in Plasmax (red boxes). Generally, after ten days in physiological medium, the culture looked more disorganized, and there seemed to be more dead cells (red arrows).

Cell viability assays using CellTiterGlo revealed that PANCO09b grown in 2D showed a slightly higher growth rate/day and lower doubling time in hours in Plasmax compared to adv.DMEM/F-12 (Figure 3D, Appendix A; *p* < 0.0001 and *p* < 0.0001, respectively). No differences in growth rate/day or doubling time were observed for PANCO11b grown in Plasmax vs. adv.DMEM/F-12.

### 3.4. Physiological Cell Culture Medium Does Not Affect Chemosensitivity of 2D Transformed Cell Cultures

Next, we compared the chemosensitivity of 2D PANCO09b and PANCO11b cell cultures growing in physiological or conventional culture medium. Dose–response curves of PANCO09b and PANCO11b to all five chemotherapeutics can be seen in Figure 4A,B. Dose–responses were reproducible in all three experiments, except for the last exposure to 5-FU. Consequently, data gained from the last 5-FU exposure were not taken into account for statistical analysis. Cell viability decreased dose-dependently in all cases when the concentration of chemotherapeutics was increased. The dose–response curves did not show any differences between Plasmax and adv.DMEM/F-12 culture media. None of the drugs reduced viability by 100% at maximal dose. *IC*_50_ values were calculated using the dose–response curves and were similar for each cell culture growing in conventional or physiological media (Figure 4C, Appendix A). GR_50_ calculations to correct for growth rate revealed no differences for PANCO09b, but PANCO11b showed higher sensitivity to SN-38 when cultured in physiological medium (Figure 4D, Appendix A
*p* = 0.0149). Thus, growth medium composition did generally not affect the drug-responsiveness of 2D transformed organoids, except for PANCO11b cells exposed to SN-38 when corrected for growth rate.

## 4. Discussion

In the current paper, we have shown that the sensitivity of PDOs to chemotherapeutics commonly used to treat pancreatic cancer is generally not affected by 2D or 3D culturing or by the composition of culture media. The apparent robustness of the chemosensitivity assay facilitates the clinical application of patient-derived primary tumor cell drug screenings by simplifying the workflow and lowering the cost and time required.

Cellular characteristics known to be relevant for drug-responsiveness, in particular proliferation rate, were not overtly affected by transformation of regular PDOs cultured in 3D into 2D monolayers, even though two cell cycle related genes (CDKL3 and MYCN) were differentially expressed in 3D vs. 2D cultures. Of note, only 29 additional genes in total were differentially expressed between 2D and 3D cultures, of which 8 were RNA- and pseudogenes. These genes are not known to play a role in the pathways affected by the used chemotherapeutics (i.e., inhibition of DNA synthesis by terminating DNA chain elongation by gemcitabine [23]; inhibition of the unwinding of DNA by inhibiting topoisomerase 1B by SN-38 [24]; targeting microtubules leading to mitotic arrest by paclitaxel [25]; terminating RNA- and DNA- synthesis by 5-FU [26]; crosslinking DNA strands leading to inhibition of RNA synthesis and transcription by oxaliplatin [27]).

Minami et al. investigated genetic, transcriptional, and morphological characteristics of nine established pancreatic cancer cell lines (PK-8, PK-45P, PK-59, PK-1, T3M-4, PANC-1, KP4, MIA PaCa-2, HPDE6) growing in 2D monolayer or as 3D spheroids. In contrast to our results, all cultures proliferated faster in 2D compared to 3D [28]. However, their approach was the opposite of ours, as they used established 2D cell cultures and converted them to 3D spheroids, whilst we established 3D PDOs that were transformed to monolayer cell cultures. Abugomaa et al. used a similar approach to ours, converting dog bladder cancer organoids growing in 3D into monolayer cell cultures that were then referred to as ‘2.5D’ cultures. In contrast to what we observed, these monolayer organoid-derived cells proliferated faster than the corresponding 3D organoids [10]. Thus, the impact of culturing cells in 2D or 3D on growth rate may be cell type specific.

After transformation of the 3D organoids to 2D cell cultures, we investigated their sensitivity to gemcitabine, paclitaxel, 5-FU, SN-38, and oxaliplatin. Both PANCO09b and PANCO11b showed no differences in chemosensitivity between the two culturing dimensions, except for SN-38, to which PANCO11b was more sensitive when cultured in 3D. So far, similar comparisons of drug responsiveness of cells cultured in 2D versus 3D in the literature were mostly done using spheroids from established cell lines. For instance, Minami et al. conducted drug-responsiveness assays to gemcitabine and 5-FU using six pancreatic cancer cell lines growing as spheroids [28]. Cell viability was measured after four days of exposure to 10 µM or 100 µM of either drug. In contrast to our results, these cell lines showed resistance to both drugs at these concentrations. Our maximal administered dose of 5-FU was 50 µM, which was sufficient to reduce cell viability to less than 50%, whilst Minami et al. observed almost 100% cell viability for all of the cell lines. Additionally, the highest administered dose of gemcitabine in the current study was 2 µM, which was sufficient to reduce cell viability to less than 2% for 2D cultures and less than 10% for 3D cultures. In contrast to these results, Melissaridou et al. observed that head and neck squamous cell carcinoma cell lines LK0902, LK0917, and LK1108 showed higher resistance to cisplatin when cultured in 3D. However, as established cell cultures are known to change their genetic profile over decades of culturing, it is likely that their chemosensitivity also changes [29]. Thus, biopsy-derived primary cell cultures are thought to be a superior model in precision medicine [30]. It is therefore interesting to note that Abugomaa et al. found similar responses to three chemotherapeutics of primary dog bladder organoid-derived 2D cultures compared to their original 3D organoid after transformation [10], which is in line with our data.

SN-38 was the only agent to which PANCO11b showed higher sensitivity when cultured in 3D. This finding cannot be explained with the common findings in the literature. In general, when differences in chemosensitivity are observed between cells cultured in 2D versus 3D, the 3D condition is more resistant [31,32]. This has been suggested to be related to cell-matrix and cell–cell interactions [31]. For example, Flörkemeier and colleagues tested a new drug (P8-D6) targeting topoisomerase in comparison to established inhibitors. They compared relative caspase 3/7 activity of breast cancer cell lines (MCF 7, SkBr3, MDA-MB231, MDA-MB468, BT-20) growing as 2D monolayer or 3D spheroid exposed to the same concentrations of chemotherapeutics. For all six cancer lines, caspase 3/7 activity was at least 10x higher when cells were cultured as 2D monolayer compared to 3D spheroids. Consequently, in contrast to our findings, the *IC*_50_ of cells cultured as 2D monolayer was lower than for the spheroids [33]. Since SN-38 acts by inhibiting topoisomerase 1B, thereby preventing DNA from unwinding [34], a potential impact of 2D versus 3D culture conditions on topoisomerase 1B activity could explain this result, although we did not observe differences at the mRNA level. Our findings suggest that the chemosensitivity of these cells is not affected by differences in penetration of the chemotherapeutic drugs between 3D and 2D cultures. This highlights that chemosensitivity might be more related to stromal factors in the tumor microenvironment. In line with this, when colorectal cancer cells were co-cultured with different types of fibroblasts, this resulted in either increased or decreased proliferation, in parallel with increased and decreased chemosensitivity, depending on the fibroblast line [35].

Next to cell–cell and cell-matrix interactions imposed by 2D vs. 3D culture environments of cells, cellular metabolic status secondary to nutrient composition and concentration in the culture medium has been reported to affect major cellular characteristics such as morphology, proliferation rate, and sensitivity to drugs [13,14,15]. In line with this, the PANCO09b cells had a higher proliferation rate and shorter doubling time in Plasmax, a novel cell culture medium with physiological nutrient composition and concentration. In addition, the Plasmax physiological cell medium changed growth properties of PANCO09b and PANCO11b cultures, increasing growth on top of other cells. This may be explained by the presence of trace elements like selenium in Plasmax that are lacking in adv.DMEM/F-12. Selenium is incorporated in selenoproteins, such as glutathione peroxidase (GPX). Higher activity of this antioxidant in Plasmax has been reported to prevent ferroptosis, a type of cell death compromising the ability of colony forming [12].

Importantly, the use of physiological culture medium instead of conventional adv.DMEM/F12 did not affect the sensitivity of PANCO09b and PANCO11b towards several clinically applied chemotherapeutic drugs. This is not in line with the data of Richards et al. [13] who showed that cells were more resistant to 5-FU because the synthesis of cytotoxic 5-FU metabolites is inhibited by the physiological uric acid concentrations present in Plasmax. However, Richards and colleagues cultured acute myeloid leukemia cells (NOMO1), which might respond differently than pancreatic cancer cells. Along this line, breast cancer cells grown in physiological medium have been shown to express higher levels of GPX [12], which may reduce oxidative stress, one of the mechanisms through which gemcitabine, paclitaxel, and oxaliplatin damage cells [36,37,38]. However, the comparable chemosensitivity in both culture media that we observed is not consistent with this concept, and may indicate that the main mechanism of action for all five drugs, interfering in DNA synthesis, is not affected by media composition for pancreatic cancer cells. Of note, it can be generally questioned whether physiological cell culture media are superior in creating a more realistic microenvironment for primary tumor cell cultures ex vivo, as solid tumors in vivo are not exposed to nutrient concentrations present in human plasma. In reality, they create their own tumor microenvironment (TME), rewire their metabolism, and deplete nutrients in the environment [39].

## 5. Conclusions

In conclusion, we have shown that various culture conditions do not majorly affect chemosensitivity of primary human pancreatic tumor cells. This indicates that assessing chemosensitivity of pancreatic cancer cell cultures is robust and reproducible. As it is more difficult to generate a personalized 2D cell culture from a biopsy than to generate a 3D organoid culture, we suggest to first establish 3D organoids and subsequently transform them into organoid-derived primary monolayer cell cultures if the aim is to test for chemosensitivity. In this way, the culture is easier to handle and more cost-efficient.

## Figures and Tables

**Figure 1 cancers-14-05617-f001:**
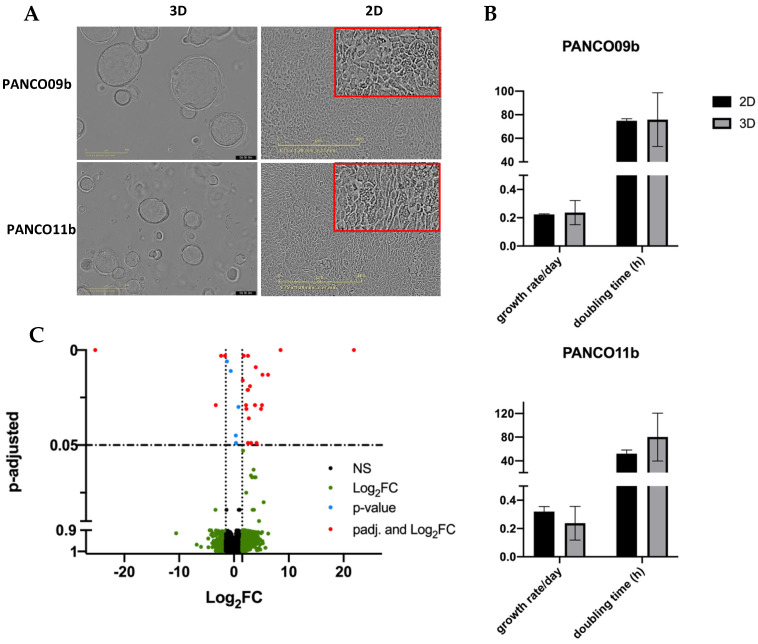
Basic characteristics of 3D and 2D cultured PANCO09b and PANCO11b cells grown in adv.DMEM/F-12. (**A**): Morphology of PANCO09b and PANCO11b grown in 3D or 2D. 4× magnification (3D) and 10× magnification (2D). (**B**): Doubling time in hours and growth rate/day of PANCO11b grown in 3D or 2D. Data are presented as mean ± SD. (**C**): Volcano plot representation of differential gene expression analysis between 3D organoids and corresponding 2D cultured cells. Red dots show significantly differentially expressed genes in 3D compared to 2D.

**Figure 2 cancers-14-05617-f002:**
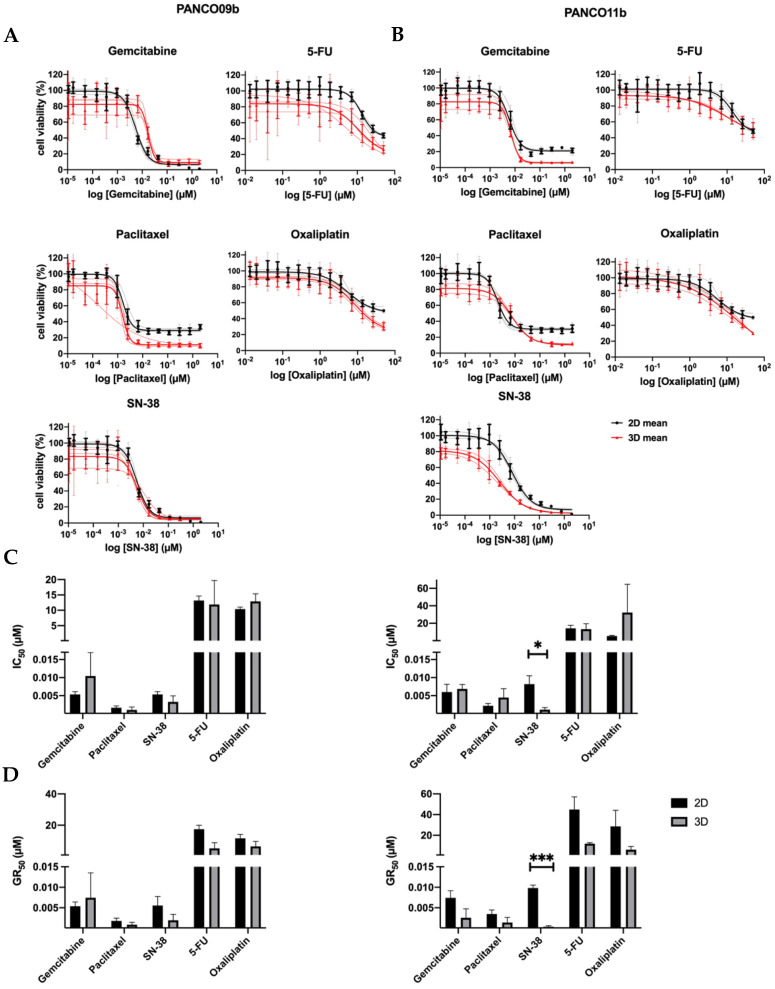
Chemosensitivity of PANCO09b and PANCO11b cells grown in adv.DMEM/F-12 in 3D vs. 2D. (**A**): Dose–response curves of PANCO09b grown in 3D vs. 2D. (**B**): Dose–response curves of PANCO11b grown in 3D vs. 2D. Thick, dark lines represent the mean and thin, light lines show the independent experiments in triplicates. (**C**): Bar graph of *IC*_50_ values of PANCO09b and PANCO11b grown in 3D vs. 2D. (**D**): Bar graph of GR_50_ values of PANCO09b and PANCO 11b grown in 3D vs. 2D. Data are presented as mean ± SD of three independent experiments performed in triplicate. Independent sample *t*-test was performed to assess differences. * = *p* < 0.05, *** = *p* < 0.001.

**Figure 3 cancers-14-05617-f003:**
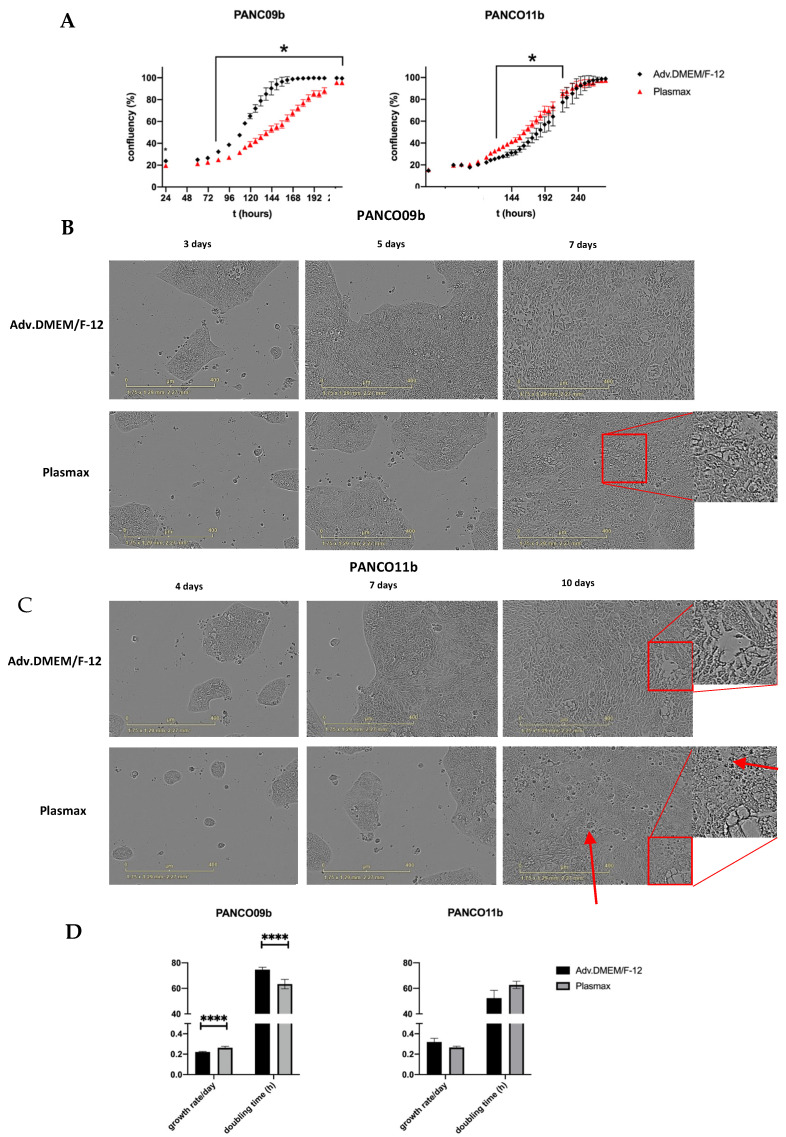
Characteristics of PANCO09b and PANCO11b cells grown in 2D in adv.DMEM/F-12 versus Plasmax. (**A**): Growth curves of 2D cultured PANCO09b and PANCO11b grown in adv.DMEM/F-12 or Plasmax medium. Images were taken every 2 h by Sartorius Incucyte S3 (not all data points shown). First shown data point is 24 h due to attaching of the cells to the surface of the plate. Data are presented as mean ± SD of one experiment performed with 6 replicates. Two-way ANOVA with Sidak’s correction for multiple comparisons was performed to analyze the effect of media over time. (**B**): Images of 2D cultured PANCO09b grown in adv.DMEM/F-12 or Plasmax after 3, 5, and 7 days. (**C**): Images of 2D cultured PANCO11b grown in adv.DMEM/F-12 or Plasmax after 4, 7, and 10 days. Images were taken by Sartorius Incucyte S3 at 10× magnification. (**D**): Doubling time in hours and growth rate/day of 2D cultured PANCO09b and PANCO11b grown in adv.DMEM/-F-12 and Plasmax. Data are presented as mean ± SD of three independent experiments performed in triplicate. Independent sample t-test was performed to assess differences. * = *p* < 0.05, **** = *p* < 0.0001.

**Figure 4 cancers-14-05617-f004:**
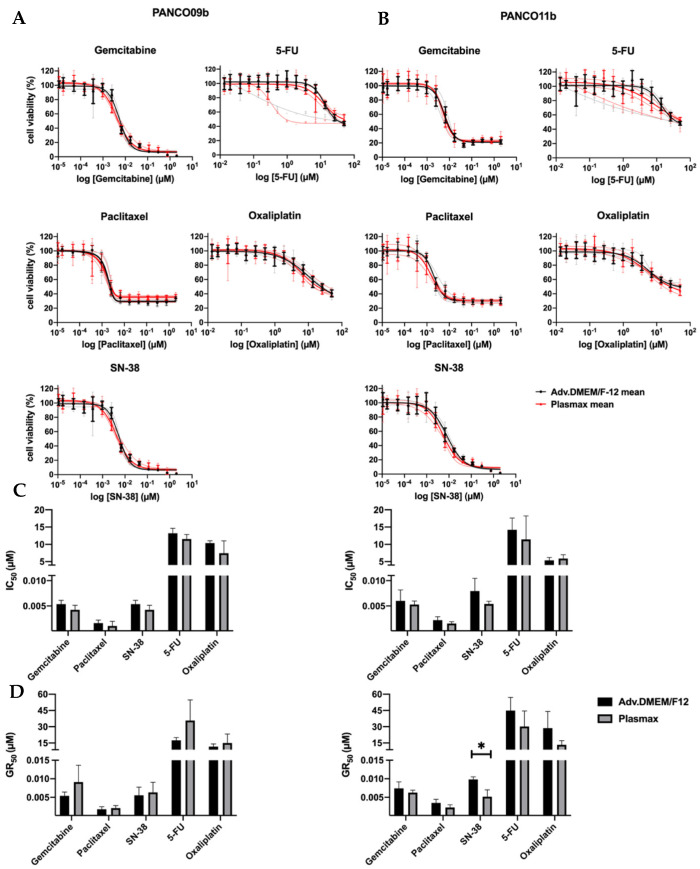
Sensitivity of 2D cultured PANCO09b and PANCO11b cells to gemcitabine, paclitaxel, SN-38, 5-FU, and oxaliplatin in adv.DMEM/F-12 versus Plasmax. (**A**): Dose–response curves of PANCO09b cells grown in adv.DMEM/F-12 or Plasmax. (**B**): Dose–response curves of PANCO11b cells grown in adv.DMEM/F-12 or Plasmax. Thick, dark lines represent the mean and light, thin lines show independent experiments in triplicates. (**C**): Bar graph of *IC*_50_ values of PANCO09b and PANCO11b cells grown in adv.DMEM/F-12 or Plasmax. (**D**): Bar graph of GR_50_ values of PANCO09b and PANCO11b cells grown in adv.DMEM/F-12 or Plasmax. Data are presented as mean ± SD of three independent experiments performed in triplicate. Independent sample *t*-test was performed to assess differences. * = *p* < 0.05.

## Data Availability

The data presented in this study are available on request from the corresponding author.

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
