# Peer review of "Chemosensitivity of 3D Pancreatic Cancer Organoids Is Not Affected by Transformation to 2D Culture or Switch to Physiological Culture Medium"

_cancers, 2022, doi:10.3390/cancers14225617_

Round 1
Reviewer 1 Report
In this highly interesting paper Gassl et al describe studies on the sensitivity of tumor derived pancreatic cancer organoids grown in 3d as well in 2 dimensional culture for chemotherapy. Furthermore they studied the influence of medium composition of the response to these chemotherapeutic agents. The study is well designed, performed and data are highly relevant for the scientific community. The fact that the 3D derived 2D cultures show similar responses to the drugs tested opens possibilities to used 2D culture for application where 3D culture would result in technical challenges. Some comments are listed below:
1. The experiments are performed with 2 individual Pancreatic cancer organoids and these do show some patient variation as expected. It would be great if additional data from additional independent organoid culture(s) would be available.
2. Was there any difference in the mutational profile of the 2 organoids used?
3. The authors show that going from 3d to 2d does not change the sensitivity profile, but have they also assessed the other war around? If the culture is established as 2D culture and in parallel in 3D, are responses also similar or is initial growth in 3D needed?
4. Did the authors every time establish new 2D lines or were they maintained in culture for longer time? Does the sensitivity change over time? I could image that the diversity of cells is lost after long time culture in 2D.
5. Does the secretome of the 3D versus 2D cells change? This given the potential important effects of the neighbouring cancer-associated fibroblasts?
6. Did the authors have the possibility to compare organoid/2D organoid responses to patient responses?
7. In 3.4 the authors conclude that the drug response is independent of the growth medium composition, but the growth is affected by the medium composition. This should be indicated.
8. It is surprising to see that apparently penetration issues do not play a major role in the sensitivity of these cells to chemotherapy regimens, indicating this might be more stroma regulated. Would be good to include this in the discussion of the manuscript.
Author Response
Response to Reviewer #1:
In this highly interesting paper Gassl et al describe studies on the sensitivity of tumor derived pancreatic cancer organoids grown in 3d as well in 2 dimensional culture for chemotherapy. Furthermore they studied the influence of medium composition of the response to these chemotherapeutic agents. The study is well designed, performed and data are highly relevant for the scientific community. The fact that the 3D derived 2D cultures show similar responses to the drugs tested opens possibilities to used 2D culture for application where 3D culture would result in technical challenges.
Response: Thank you very much for the appreciation of our study.
Some comments are listed below:
- Comment: The experiments are performed with 2 individual Pancreatic cancer organoids and these do show some patient variation as expected. It would be great if additional data from additional independent organoid culture(s) would be available.
Answer: It is indeed true that we observed variation in drug responses between the two patient-derived organoids that we tested, which is in line with the broad literature on organoid drug screens. Since patient variation is known to exist and already widely studied, our aim for the current paper was not to show differences between organoid cultures derived from different patients. Instead, we focused on showing that for a given organoid culture, chemosensitivity does not significantly change when the culture is grown in 3D versus 2D, or when grown in conventional versus physiological media. We believe that the data that we generated with two independent organoid cultures are novel and consistent. Nevertheless, we agree with the reviewer that additional 2D data would be of interest, and have meanwhile generated more 2D cell cultures from 3D patient-derived organoid cultures, showing that it is feasibly to expand our approach. However, since we have not conducted additional drug screens with these novel 2D cultures yet, we have not included these data in the manuscript.
- Comment: Was there any difference in the mutational profile of the 2 organoids used?
Answer: The mutational profile of these 2 organoid cultures has been previously investigated by our group and was published by Vaes et al. in 2020 in the Journal of Cachexia, Sarcopenia and Muscle [1]. In summary, PANCO09b had a homozygous missense KRAS mutation (G12V) and a homozygous missense TP53 mutation (S215I). PANCO11b had a heterozygous missense KRAS mutation (G12R) and a homozygous missense TP53 mutation (Y205H).
- Comment: The authors show that going from 3D to 2D does not change the sensitivity profile, but have they also assessed the other way around? If the culture is established as 2D culture and in parallel in 3D, are responses also similar or is initial growth in 3D needed?
Answer: We agree with the reviewer that this is an interesting research approach from a scientific point of view. However, in our and other’s [2] experience, it is much harder to directly establish 2D cultures from tumor tissue digests as compared to 3D organoid cultures, which can be established with >90% efficiency in our lab. Consequently, we consider that the currently reported approach in which 3D cultures are converted to 2D cultures is more in line with the timeline required for clinical translation of primary tumor cell culture drug screens, and therefore more relevant to disseminate. Since cellular characteristics such as gene expression profiles and proliferation rate were not majorly different between the 2D and 3D cultures, we principally don’t expect that it will make a difference to go from 2D to 3D or from 3D to 2D.
- Comment: Did the authors every time establish new 2D lines or were they maintained in culture for longer time? Does the sensitivity change over time? I could image that the diversity of cells is lost after long time culture in 2D.
Answer: We have not established a new independent 2D line for each assay reported in the current paper, but rather maintained a once converted 2D culture for a longer time (also including freeze/thaw cycles). However, as we have performed several independent experiments, the last chemosensitivity test was at a considerably higher passage number than the first. Importantly, no systematic shift in sensitivity was observed for either organoid culture in either 2D or 3D conditions. The reviewer is right that some cellular heterogeneity may be lost after a longer period of culturing. This, however, also likely happens in 3D cultures, as faster proliferating cells will accumulate over time also in 3D. To overcome this problem, or reduce its impact, we have conducted all experiments under passage 20.
- Comment: Does the secretome of the 3D versus 2D cells change? This given the potential important effects of the neighbouring cancer-associated fibroblasts?
Answer: Whereas we agree that investigating the impact of 3D versus 2D culturing on the secretome of the cells is interesting by itself, we consider that showing this goes beyond the scope of the current paper. Our research question focused on the drug responsiveness of organoid cultures cultured in different conditions. Nevertheless, given that there were no dramatic effects of 2D versus 3D culturing on gene expression of the cells, we don’t expect that the secretome will be significantly impacted by 2D or 3D culturing. The presence or absence of CAFs is apparently no major factor in this respect.
- Comment: Did the authors have the possibility to compare organoid/2D organoid responses to patient responses?
Answer: Thank you very much for this interesting and important question. The predictive power of organoid-based drug screens has been recently evaluated by Wensink et al. [3]. Out of 17 studies using organoids, 5 reported a significant correlation between organoid and patient response, and 11 reported a positive trend, highlighting the potential of organoid drug screens in general. We would therefore expect to find a comparably similar congruency between the response of the organoids and the corresponding patients that we studied here. However, unfortunately, we only have the treatment data of patient 9 (donor of PANCO09b), who was treated with gemcitabine and paclitaxel after local recurrence. The patient survived for 87 weeks (20 months). For patient 11 (donor of PANCO11b), we only know that the patient survived for 45 weeks (10 months). Unfortunately, the patient was referred to another hospital, and we do not know whether any chemotherapeutics were taken. We can only compare organoid and patient responses relatively if data for patient 11 would have been available.
- Comment: In 3.4 the authors conclude that the drug response is independent of the growth medium composition, but the growth is affected by the medium composition. This should be indicated.
Answer: We agree with the reviewer and have therefore adjusted the text:
‘Thus, growth medium composition did generally not affect the drug-responsiveness of 2D transformed organoids, except for PANCO11b cells exposed to SN-38 when corrected for growth speed.’
- Comment: It is surprising to see that apparently penetration issues do not play a major role in the sensitivity of these cells to chemotherapy regimens, indicating this might be more stroma regulated. Would be good to include this in the discussion of the manuscript.
Answer: We agree with the reviewer that this is a remarkable and unexpected implication of our data. We have there expanded the discussion section as follows:
‘Our findings suggest that the chemosensitivity of these cells is not affected by differences in penetration of the chemotherapeutic drugs between 3D and 2D cultures. This highlights that chemosensitivity might be more related to stromal factors in the tumor microenvironment. In line with this, when colorectal cancer cells were co-cultured with different types of fibroblasts, this resulted in either increased or decreased proliferation, in parallel with increased and decreased chemosensitivity, depending on the fibroblast line [4].’
References
[1] Vaes, R. D. W.; van Dijk, D. P. J.; Welbers, T. T. J.; Blok, M. J.; Aberle, M. R.; Heij, L.; Boj, S. F.; Olde Damink, S. W. M.; Rensen, S. S. Generation and Initial Characterization of Novel Tumour Organoid Models to Study Human Pancreatic Cancer-Induced Cachexia. J. Cachexia. Sarcopenia Muscle 2020, 11 (6), 1509–1524.
[2] Baker, L. A.; Tiriac, H.; Tuveson, D. A. Generation and Culture of Human Pancreatic Ductal Adenocarcinoma Organoids from Resected Tumor Specimens. In Methods in Molecular Biology; Methods Mol Biol, 2019; Vol. 1882, pp 97–115.
[3] Wensink, G. E.; Elias, S. G.; Mullenders, J.; Koopman, M.; Boj, S. F.; Kranenburg, O. W.; Roodhart, J. M. L. Patient-Derived Organoids as a Predictive Biomarker for Treatment Response in Cancer Patients. npj Precision Oncology. Nature Publishing Group April 12, 2021, pp 1–13.
[4] Koh, B.; Jeon, H.; Kim, D.; Kang, D.; Kim, K. R. Effect of Fibroblast Co-Culture on the Proliferation, Viability and Drug Response of Colon Cancer Cells. Oncol. Lett. 2019, 17 (2), 2409–2417.
Reviewer 2 Report
1. Did the authors perform any histological analysis like H&E staining etc. to compare the pathological differences or similarities between the 3D, and 2D cultures and original patient tumors? Did the authors perform any immunofluorescent staining for understanding the expression of proteins in 3D vs 2D? Without these analyses, it is difficult to say if the 2D cultures truly represent the patient's tumors.
2. Did the authors compare the gene expression profile of the primary tumor vs tumor organoid vs 2D cultures? Or can they comment on how that would look? Without a comparison of all these cultures, the entire picture is not clear to the readers.
3. Can the authors comment on why PANCO09b 2D and 3D cultures showed similar IC50 values?
4. Can the authors comment on how the IC50 or growth rates would be affected if there were co-cultures of cancer cells and stromal cells as compared to only cancer cells?
Round 2
Reviewer 1 Report
All comments have been carefully addressed!
Reviewer 2 Report
The authors have addressed all my comments